# EXPLICIT COLUMN RELATIONSHIP-BASED DIFFUSION MODEL FOR HIGH-QUALITY SYNTHETIC TABULAR DATA GENERATION

## ABSTRACT

Tabular data plays a vital role in critical applications such as healthcare, finance, and education. Its effective utilization in data-driven models is frequently hindered by data scarcity and privacy concerns. In response, synthetic tabular data generation has emerged as a powerful solution that provides privacy-preserving data mirroring real-world distributions. However, many existing generative models still struggle to preserve the complex column relationships within tabular data. Additionally, they often fail to account for the real-world constraints that are essential for ensuring the authenticity and practical usability of the generated data. In this paper, we propose ECR-DM, the **E**xplicit **C**olumn **R**elationship-Based **D**iffusion **M**odel for synthetic tabular data generation. In the forward diffusion process, we introduce the Noise Perturbation Mechanism, which enables the model to learn column distributions in a fine-grained manner. In the reverse diffusion process, we incorporate Constraint-Guided Recovery, which guides the model to recover inter-column dependencies and restore the true data distribution. NPM helps the diffusion model capture the detailed column-wise characteristics of the data, while CGR ensures the preservation of inter-column relationships and the high-quality synthetic tabular data generation. We validate the effectiveness of our approach through extensive experiments on six tabular data benchmarks. Our model outperforms state-of-the-art methods across seven evaluation metrics, particularly in downstream tasks. Code is available at https://anonymous.4open.science/r/ECR-DM-0C72.

## 1 INTRODUCTION

Tabular data is a fundamental format in diverse domains such as finance (Sattarov et al., 2023), healthcare (He et al., 2024), and education (Borisov et al., 2022a), where accurate decision-making and reliable predictions critically depend on high-quality data. However, its usability is often limited due to challenges re-

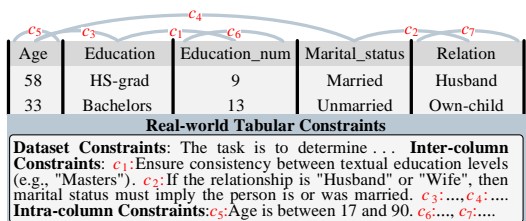

Figure 1: Illustration of Complex Column Relationships and Real-world Tabular Constraints of Adult dataset.

lated to data privacy protocols (Regulation, 2018; Illman & Temple, 2019) and data quality issues (e.g., missing values (Zhang et al., 2024), imbalanced data (Kim et al., 2024), and limited data volume (Seedat et al., 2023)). These challenges reduce data usability and hinder the development and deployment of predictive models (Zhang et al., 2024; Shi et al., 2025a).

Synthetic tabular data generation (Shi et al., 2025b) has emerged as a promising solution to address challenges like data privacy (Jordon et al., 2018; Zhao et al., 2024), quality (Kim et al., 2022b), and availability (Kim et al., 2022a). In recent years, methods for synthetic tabular data generation have advanced rapidly, yet significant challenges remain. Early approaches, such as the VAE-based method (Xu et al., 2019; Ma et al., 2020; Liu et al., 2023) and GAN-based method (Choi et al., 2017; Lee et al., 2021; Zhao et al., 2021), focus on capturing the marginal and joint distributions of columns. However, these methods fail to model complex tabular structures. They focus primarily on distributions and overlook the real, intricate relationships between columns (as shown in the top part of Fig. 1), not to mention the challenge of incorporating real-world tabular constraints. **How**

**to construct a model that captures the complex structures of tabular data while adhering to real-world constraints is crucial for synthetic tabular data generation.**

More recently, diffusion-based methods have led to significant advancements in high-quality synthetic tabular data generation by learning through progressive denoising. For example, TabDDPM (Kotelnikov et al., 2023) and TABDIFF (Shi et al., 2025a) combine continuous (Song et al., 2020; Karras et al., 2022) and discrete (Austin et al., 2021) diffusion to model numerical and categorical column distributions. CSDI (Zheng & Charoenphakdee, 2022) and TABSYN (Zhang et al., 2024) encode tabular data into a latent space to model the overall data distribution. While these methods show great promise, they still focus primarily on data distribution and fail to capture the complex **inter-column dependencies** (Shi et al., 2025b) that are essential for generating realistic synthetic tabular data. Furthermore, they often overlook **real-world constraints** (e.g., Real-world Tabular Constraints of Fig. 1) during the generation process, resulting in synthetic data that may violate domain-specific rules and fail to align with actual data patterns. Due to the unique reconstruction process of diffusion models, they can incorporate realistic constraints, giving them a distinct advantage in this field. Therefore, building upon diffusion models, we can effectively capture column relationships and generate data that adheres to real-world constraints.

To address the above limitations, we propose ECR-DM, a novel **E**xplicit **C**olumn **R**elationship-Based **D**iffusion **M**odel for synthetic tabular data generation. Our model is designed to explicitly capture the complex column relationships while adhering to real-world constraints. The key innovation of ECR-DM is to capture finer-grained column distributions through the **Column-level Forward Process** and to recover inter-column dependencies through the **Tabular Constraint-guided Reverse Process**. In the Column-level Forward Process, we introduce a **Noise Perturbation Mechanism** (NPM), which adds varying levels of noise to each column individually. This process enables the model to learn column-wise characteristics at a fine-grained level, enhancing its ability to capture the intricate relationships between individual columns. The column-specific noise perturbation also facilitates the modeling of complex column distributions, which are essential for understanding the underlying data structure. This step is crucial for training the model to differentiate subtle variations in the data, thus improving the model's capacity to identify and preserve inter-column dependencies during the generation process. In the Tabular Constraint-guided Reverse Process, we incorporate **Constraint-Guided Recovery** (CGR), which guides the model in the denoising process by enforcing real-world tabular constraints. These constraints are based on domain-specific rules and logical dependencies between columns, ensuring that the synthetic data respects the inherent relationships in the real-world data. By integrating these constraints, the model is able to effectively restore inter-column dependencies, preserving the structural integrity of the original data. This step is essential for reconstructing the data distribution from a column-wise perspective, ensuring that the generated synthetic data not only follows statistical distributions but also aligns with real-world patterns and business rules. Through this process, the model generates high-quality synthetic tabular data that is both realistic and usable for downstream applications, such as training predictive models or conducting data analysis. Our main **contributions** are summarized as follows:

⋆ We propose the Explicit Column Relationship-Based Diffusion Model (ECR-DM), which explicitly captures complex inter-column dependencies and enforces real-world constraints during synthetic tabular data generation.
⋆ We propose NPM and CGR for the forward and reverse processes of ECR-DM, respectively. NPM adds column-specific noise, enabling fine-grained learning of column distributions and facilitating the capture of inter-column dependencies. CGR incorporates real-world constraints to restore inter-column dependencies and generate realistic, high-quality synthetic data.
⋆ We conduct extensive experiments on six benchmark tabular datasets, demonstrating that ECR-DM outperforms state-of-the-art methods across seven evaluation metrics, with particularly strong performance on downstream tasks.

## 2 RELATED WORK

The importance and scarcity of tabular data have led to growing interest in methods for synthetic tabular data generation. Early approaches, such as TVAE (Xu et al., 2019) and CTGAN (Xu et al., 2019), applied VAEs (Kingma et al., 2013) and GANs (Goodfellow et al., 2014) to tabular data, effectively modeling the marginal and joint distributions of columns. However, due to limitations in model capacity, these methods struggled to capture the complex structure of tabular data, limiting their generative capabilities. In recent years, the introduction of physical diffusion processes

has significantly advanced data generation, leading to breakthroughs in areas such as image generation (Rombach et al., 2022), text-to-image generation (Saharia et al., 2022), and video generation (Ho et al., 2022). Motivated by these successes, diffusion models have increasingly been applied to synthetic tabular data generation. Models like TabDDPM (Kotelnikov et al., 2023) and TABDIFF (Shi et al., 2025a) apply discrete diffusion (Austin et al., 2021) for categorical columns and continuous diffusion (Song et al., 2020; Karras et al., 2022) for numerical columns, capturing the distribution of both data types from a multimodal perspective. In contrast, CSDI (Zheng & Charoenphakdee, 2022) and TABSYN (Zhang et al., 2024) map both categorical and numerical data into a shared latent space, learning the data distribution within this space. There are also LLM-based methods, such as GReaT (Borisov et al., 2022b) and P-TA (Yang et al., 2024), which fine-tune LLMs by serializing tabular samples into text, allowing the model to learn the text-level distribution of tabular data. Although these models have made significant strides in synthetic tabular data generation, most existing methods primarily focus on improving distributional fidelity, neglecting a crucial aspect: high-quality synthetic data must not only match the real data distribution but also preserve inter-column dependencies and adhere to real-world tabular constraints. This limitation hinders the practical usability of tabular data generated by current methods.

## 3 Problem Definition

**Synthetic Tabular Data Generation:** Given a tabular dataset $T = \{D, C, F\}$, where $D$ denotes the set of samples, $C$ represents the corresponding real-world tabular constraints, and $F$ contains the column names of the tabular data. Each sample $x \in D$ is represented as $\{x_i\}_{i=1}^n$, where $n$ is the number of tabular columns that compose of $N_n$ numerical columns and $N_c$ categorical columns. Real-world tabular constraints $C$ include dataset constraints, intra-column constraints, and inter-column constraints, which define the domain-specific rules and data patterns that tabular data must satisfy. These constraints can typically be derived from the descriptive information provided with the tabular data. The column names are denoted as $F = \{f_i\}_{i=1}^n$. Formally, the goal of synthetic tabular data generation is to learn a parameterized generative model $g_\theta(D)$ that captures the real data distribution $p(x)$ and generates synthetic data $x' \in D'$ by sampling from it. To ensure the practicality of synthetic tabular data $D'$, it must not only conform to the real data distribution $p(x)$ but also preserve inter-column dependencies and adhere to the real-world constraints included in $C$.

## 4 Explicit Column Relationship-Based Diffusion Model

Building on the challenges outlined in the introduction, we propose ECR-DM, a method designed to explicitly capture inter-column dependencies and enforce real-world tabular constraints for synthetic tabular data generation, as shown in Fig. 2.

### 4.1 Column-level Forward Process

The Column-level Forward Process aims to overcome the challenges of modeling complex dependencies between columns in tabular data. Unlike traditional forward processes, which apply noise uniformly across the entire sample. The NPM applies column-specific noise to each column. This design enables the model to capture the fine-grained distributions of each column, allowing it to learn inter-column dependencies more effectively. To model inter-column dependencies effectively, we first transform the heterogeneous columns into a unified semantic space.

### 4.1.1 Unified Semantic Space

Tabular data consists of heterogeneous columns, including categorical and numerical features, each with its own characteristics and distribution. To effectively capture the relationships between these columns, we encode the data into a unified semantic space $\mathbb{R}^d$ using a Pre-trained Language Model (PLM), such as BERT (Devlin et al., 2019). This approach allows the model to process both categorical and numerical columns in a shared representation space.

For **categorical columns**, we combine the feature name $f_i$ and its corresponding value $x_i$ and pass them through the PLM. The resulting embedding is averaged to obtain the column's semantic representation. For **numerical columns**, we combine the feature name $f_i$ with the value $x_i$ via multiplication, preserving the numerical nature of the column. The encoding process is as follows:

$$h_i = \begin{cases} \text{Average}(\text{PLM}([f_i, x_i])), & \text{for Categorical Columns,} \\ \text{Average}(\text{PLM}([f_i])) \times x_i, & \text{for Numerical Columns,} \end{cases} \tag{1}$$

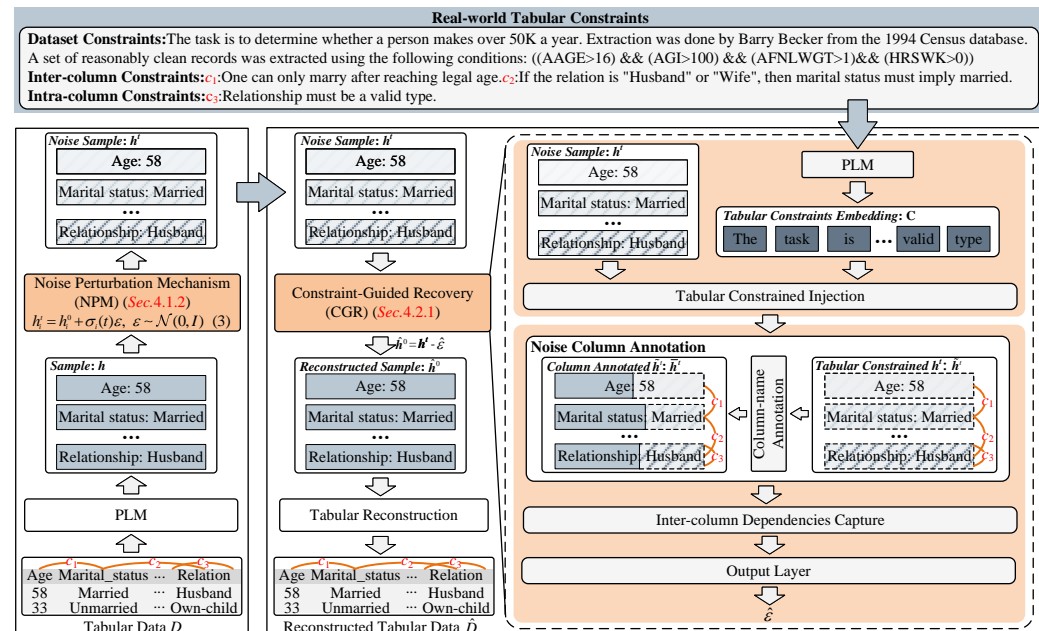

Figure 2: An overview of ECR-DM. ECR-DM consists of two main processes: the Column-level Forward Process, which introduces a NPM to learn column distributions and facilitate dependencies capture; the Tabular Constraint-guided Reverse Process, which uses CGR to integrate real-world constraints and restore inter-column dependencies.

where $f_i$ is the column name (e.g., Age) and $x_i$ is the corresponding value (e.g., 58). This approach projects both categorical and numerical columns into the same semantic space, facilitating the learning of their dependencies. After encoding tabular data into a unified semantic space, we apply Noise Perturbation Mechanism to model column-specific dependencies and fine-grained distributions.

### 4.1.2 NOISE PERTURBATION MECHANISM

The general Forward Diffusion Process has been widely used in generative models to match the data distribution $p(x)$ by adding noise uniformly to the entire data sample. However, this approach does not capture the fine-grained dependencies between individual columns in tabular data, which are essential for generating realistic data. Specifically, while the forward process can model the overall data distribution, it fails to refine the distribution of individual columns $p_i(x_i)$, leading to potential conflicts between columns. For example, the model might generate plausible data overall, but fail to capture the inter-column relationships, such as mismatches between categorical features (e.g., the sex column as "men" and the relationship column as "wife").

To address this limitation, we introduce the NPM in the Column-level Forward Process, where noise is applied individually to each column. This enables the model to learn the distribution of each column more effectively, while also preserving the inter-column dependencies. The column-specific noise perturbation allows the model to capture dependencies at a finer granularity, ensuring more realistic data generation. Formally, the forward diffusion process with NPM is described as:

$$h_i^t = h_i^0 + \sigma_i(t)\varepsilon, \quad \varepsilon \sim \mathcal{N}(0, \mathbf{I}), \tag{2}$$

where $h_i^t$ is the diffused embedding of the $i$-th column at time $t$, $\sigma_i(t)$ is the noise level applied to the $i$-th column, and $h_i^0$ is the initial representation of the column obtained from the PLM. This column-level perturbation ensures that the model captures the fine-grained distribution $p_i(x_i)$ or each column, improving its ability to model inter-column dependencies.

While this design facilitates learning from individual column distributions, it also introduces additional complexity in the denoising process. To maintain consistency with the original forward diffusion process, we extend it to operate at the column level in the reverse process:

$$dh_i^t = -2\dot{\sigma}_i(t)\sigma_i(t)\nabla_{h_i^t} \log p(h_i^t)\,dt + \sqrt{2\dot{\sigma}_i(t)\sigma_i(t)}\,d\omega_t, \tag{3}$$

where $\sigma_i(t)$ is the noise level of $i$-th column and $\omega_t$ is the standard Wiener process. This column-level reverse diffusion process allows the model to reconstruct data that reflects real-world pat-

terns by utilizing finer-grained column distributions. However, adding varying noise levels across columns increases the complexity of the denoising task. This challenge will be addressed in the next step: the Tabular Constraint-guided Reverse Process, which incorporates real-world tabular constraints to guide the model in recovering the original data distribution and improving the quality of the generated data.

## 4.2 TABULAR CONSTRAINT-GUIDED REVERSE PROCESS

The Tabular Constraint-guided Reverse Process aims to recover the original tabular data distribution and preserve inter-column dependencies while ensuring that the generated data adheres to real-world tabular constraints. The process consists of two key components: Constraint-guided Recovery (CGR) and Tabular Reconstruction. In this section, we focus on the CGR, which incorporates Tabular Constrained Information Injection and Explicit Inter-column Dependency Capture.

### 4.2.1 CONSTRAINT-GUIDED RECOVERY

The CGR process modifies the standard reverse diffusion process by conditioning the denoising score function on real-world tabular constraints $C$, column names $f_i$, and inter-column dependencies $h_i^t$. This allows the model to recover column distributions that not only reflect real-world patterns but also preserve the relationships between columns. The score function that guides the denoising process is defined as:

$$\nabla_{h_i^t} \log p(h_i^t \mid C, f_i, h_1^t, \ldots, h_n^t). \tag{4}$$

where $h_i^t$ is the embedding of the $i$-th column at time $t$, and $C$ represents the real-world constraints (such as domain knowledge, intra-column, and inter-column dependencies). The score function is now conditioned on the constraints and column-specific information, allowing the model to explicitly capture the relationships between columns. When this score function is substituted into the reverse diffusion process (Eq. 3), we get the CGR equation:

$$dh_i^t = -2\dot{\sigma}_i(t)\sigma_i(t)\nabla_{h_i^t} \log p(h_i^t|C, f_i, h_1^t, ..., h_n^t) \, dt + \sqrt{2\dot{\sigma}_i(t)\sigma_i(t)} \, d\omega_t, \tag{5}$$

where $\sigma_i(t)$ is the noise level of $i$-th column, $\omega_t$ is the standard Wiener process. This equation governs how the column-specific embeddings are updated in each step of the reverse process, incorporating both the noise and real-world constraints. To ensure that the model can handle constraints with flexibility, we introduce a probabilistic approach to incorporate $C$ into the reverse process:

$$C = \begin{cases} C, & 1 - p_u \\ \emptyset, & p_u \end{cases}, \tag{6}$$

where $p_u$ is the probability of omitting constraints. This approach enables the model to generate data either with constraints ($\log p(h_i^t|C, f_i, h_1^t, ..., h_n^t)$) or without constraints ($\log p(h_i^t|\emptyset, f_i, h_1^t, ..., h_n^t)$), depending on the context. This flexibility allows the model to adapt to varying levels of constraint information during the data generation process.

Since the real data distribution $p(x)$ is unknown, its gradients cannot be directly computed. In other words, the score function $\nabla_{h_i^t} \log p(h_i^t \mid C, f_i, h_1^t, \ldots, h_n^t)$ in Eq. 4 is unavailable. To address this, we introduce a **Constraint-Guided Recovery Model** $s_\theta$, which approximates the Gaussian noise added to each column feature $h_i$. This allows us to approximate the score function as:

$$\nabla_{h_i^t} \log p(h_i^t \mid C, f_i, h_1^t, \ldots, h_n^t) \approx -s_\theta(h_i^t, t, C, f_i, h_1^t, ..., h_n^t)/\sigma_i(t). \tag{7}$$

This model is an integral part of the Constraint-Guided Recovery process, which works to refine the tabular data generation by capturing inter-column dependencies and real-world constraints. Specifically, the Constraint-Guided Recovery Model consists of two key components: Tabular Constrained Information Injection and Explicit Inter-column Dependency Capture.

**Tabular Constrained Information Injection.** Real-world tabular constraints $C$ provide critical domain-specific knowledge that guides the generation of synthetic data. These constraints ensure that the generated data follows real-world patterns by enforcing dependencies within and across columns. As shown in Table 1,

Table 1: Real-World Tabular Constraints $C$.

| Dataset Constraints |
| --- |
| *The dataset consists of feature vectors from 12,330 sessions, with each session belonging to...* |
| Intra-Column Constraints |
| *AGE: between 18 and 100 years. ...* |
| Inter-Column Constraints |
| *If a person is 20 years or younger, their credit limit (LIMIT_BAL) must be zero. ...* |

real-world constraints can be categorized into three types: Dataset Constraints, Intra-column Constraints, and Inter-column Constraints. To effectively incorporate these constraints into the data generation process, we utilize Tabular Constrained Information Injection. First, we encode the constraints using a PLM. After encoding, the constraints are injected into each column's noisy embedding using a cross-attention mechanism, which helps align the generated data with the real-world constraints. The injection process is mathematically represented as follows:

$$\mathbf{C} = \text{PLM}(C), \tilde{h}_i^t = \text{Softmax} \quad \left( \frac{(h_i^t W^Q) \cdot (\mathbf{C} W^K)^\top}{\sqrt{d}} \right) \cdot \mathbf{C} W^V, \tag{8}$$

where $W^Q \in \mathbb{R}^{d \times d}, W^K \in \mathbb{R}^{d \times d}$ and $W^V \in \mathbb{R}^{d \times d}$ are learnable parameters, $\mathbf{C} \in \mathbb{R}^{k \times d}$, $k$ is the number of token in real-world tabular constraints $C$. However, Tabular Constrained Information Injection alone is not sufficient to fully capture the relationships between columns. To explicitly model these inter-column dependencies, we introduce Explicit Inter-column Dependency Capture. This step is essential for ensuring that the generated synthetic data maintains realistic relationships across different columns, respecting the logical constraints that exist between them.

**Explicitly Inter-column Dependency Capture.** In the Column-level Forward Process, injecting noise at different levels to individual columns disrupts the information within each column. Additionally, the column-exchange invariance of tabular data complicates column identification, making it difficult to capture inter-column dependencies. To address this, we annotate noisy column embeddings $\tilde{h}_i^t$ with their corresponding column names $f_i$ allowing the model to leverage these annotations to better capture inter-column dependencies while maintaining intra-column distributions:

$$c_i = \text{Average}(\text{PLM}(f_i)), \overline{h}_i^t = [\tilde{h}_i^t || c_i], \tag{9}$$

where $c_i$ is the embedding of the column name $f_i$, and $[\cdot || \cdot]$ denotes concatenation. Next, we use a Dependencies Capture Model (a Transformer without positional embeddings) to model the inter-column dependencies guided by injected real-world constraints:

$$\hat{h}_1^t, ..., \hat{h}_n^t = \text{Dependencies Capture}(\overline{h}_1^t, ..., \overline{h}_n^t), \tag{10}$$

Then, we employ an MLP in the output layer to predict the noise introduced during the Column-level Forward Process, enabling the reconstruction of the corrupted $i$-th column feature $h_i^t$:

$$\hat{\epsilon}_i = \text{MLP}(\hat{h}_i^t), \qquad \hat{h}_i^0 = h_i^t - \hat{\epsilon}_i, \tag{11}$$

where $\hat{\epsilon}_i$ denotes the predict noise of the $i$-th column feature in the sample $x$, $\hat{h}_i^0$ denotes the reconstructed $i$-th column feature representation of $\hat{h}_i^t$ after the denoising process. Finally, we train the model via denoising score matching (Karras et al., 2022):

$$\mathcal{L}_d = \sum_{i=1}^{n} \|\hat{\epsilon}_i - \epsilon_i\|_2^2, \tag{12}$$

where $\epsilon_i$ is the noise added to the $i$-th column feature during the Column-level Forward Process.

### 4.2.2 TABULAR RECONSTRUCTION

After the denoising process, we obtain the denoised column embeddings $\hat{h}_1^0, \hat{h}_2^0, ..., \hat{h}_n^0$, which represent the reconstructed features of the tabular data. The goal of the Tabular Reconstruction process is to convert these embeddings back into meaningful tabular data that adheres to the original data distribution. The reconstruction process is carried out by decoding each column embedding $\hat{h}_i^0$ into the final feature values $\hat{x}_i$ using the following formulas:

$$\hat{x}_i = \begin{cases} \text{Softmax}(\hat{h}_i^0 \cdot w_i^{cat} + b_i^{cat}), & \text{for Cat}, \\ \hat{h}_i^0 \cdot w_i^{num} + b_i^{num}, & \text{for Num}, \end{cases} \tag{13}$$

where $w_i^{cat} \in \mathbb{R}^{d \times C_i}$, $b_i^{cat} \in \mathbb{R}^{1 \times 1}$, $w_i^{num} \in \mathbb{R}^{d \times 1}$, $b_i^{num} \in \mathbb{R}^{1 \times 1}$ are detokenizer's parameters[1] for categorical and numerical columns, respectively. This decoding process ensures that each column's feature values are restored in a manner that reflects both the individual column distributions

---

[1]The detokenizer model is a pretrained model, detailed in the paper Zhang et al. (2024).

and inter-column relationships. The final reconstructed tabular $\hat{x}$ is composed of all the columns $[\hat{x}_1, ..., \hat{x}_n]$. The detailed training and synthetic tabular data generation algorithm of ECR-DM is described in Appendix B.

After the tabular data is reconstructed, the quality of the generated synthetic data is evaluated using Sample Accuracy (SA), a novel metric designed to measure how well the generated data adheres to real-world tabular constraints. Unlike existing methods, SA quantifies the proportion of generated data that satisfies these constraints, ensuring the realism and usability of the synthetic data. Formally, SA is defined as: $\text{SA} = \frac{\hat{D}_{corr}}{\hat{D}}$, where $\hat{D}$ denotes the generated dataset, and $\hat{D}_{corr}$ represents the subset of generated data that violates real-word tabular constraints. SA helps assess the fidelity of the generated data, ensuring that it aligns with the logical and domain-specific rules, which is crucial for practical applications in areas such as healthcare, finance, and education.

## 5 EXPERIMENTS

To thoroughly evaluate the performance of ECR-DM, we focus on addressing three key research questions: **Q1:** To what extent does ECR-DM uphold the integrity of real-world tabular constraints during the synthetic data generation process? **Q2:** How effectively does ECR-DM replicate the distribution of real-world data while capturing inter-column dependencies? **Q3:** What is the contribution of each individual component in ECR-DM to the overall model performance?

### 5.1 EXPERMENTAL SETUPS

**Datasets.** We evaluate ECR-DM using six real-world datasets from Zhang et al. (2024), including four classification datasets (i.e., Adult, Default, Shoppers, and Magic) and two regression datasets (i.e., Beijing and News). Each dataset provides real-world tabular constraints $C$ containing data constraints and domain knowledge. Refer to Appendix C.1 for details of the dataset.

**Baselines.** We compare ECR-DM against seven baseline models that are categorized into three groups: 1) Traditional methods: CTGAN (Xu et al., 2019), CTGAN+ (Zhao et al., 2024), and TVAE (Xu et al., 2019); 2) LLM-based methods: P-TA (Yang et al., 2024); 3) Diffusion model methods: TabDDPM (Kotelnikov et al., 2023), TABSYN (Zhang et al., 2024) and TABDIFF (Shi et al., 2025a). Refer to Appendix C.2 for details of the baseline methods.

**Implementation Details.** We compare ECR-DM to all baseline methods using the same experimental setup. All the methods are optimized with Adam optimizer, and all the experiments are conducted on an Nvidia L40 GPU (48GB) with the same seed set. Our method uses a pretrained BERT-base-uncased (Devlin et al., 2019) to encode tabular data into a unified semantic space $\mathbb{R}^d$, where $d = 128$. The batch size is set to 4096, and the learning rate is set to 1e-3.

**Evaluation Methods.** We use seven metrics to evaluate the quality of the generated data in terms of authenticity and fidelity. Authenticity is assessed by Sample Accuracy (SA) and Machine Learning Efficiency (MLE, e.g., RMSE and AUC), while fidelity is evaluated using Shape, Trend, $\alpha$-Precision, $\beta$-Recall, and C2ST. A detailed description of these metrics is provided in Appendix C.3.

### 5.2 MAIN RESULTS

#### 5.2.1 AUTHENTICITY EVALUATION

To evaluate ECR-DM in terms of preserving real-world tabular constraints (Q1), we conduct experiments in Table 2 with the following two key observations.

**Diffusion-based Methods Outperform Other Approaches.** As shown in Table 2, diffusion-based methods significantly outperform traditional and LLM-based models, particularly in SA. This is especially notable for ECR-DM, which achieves the highest or near-optimal SA scores in multiple datasets. The strong performance of diffusion-based models can be attributed to their progressive denoising process. This process allows diffusion models to more effectively preserve real-world constraints and generate more realistic synthetic data, as demonstrated by the superior SA scores in comparison with baseline methods.

**ECR-DM Generates More Realistic Data.** As shown in Table 2, ECR-DM not only excels in SA scores but also significantly narrows the performance gap with real datasets on downstream tasks, such as RMSE and AUC. This is achieved by ECR-DM' ability to enforce inter-column dependencies and preserve real-world tabular constraints, ensuring that the generated synthetic data reflects realistic patterns. By incorporating these constraints directly into the model, ECR-DM produces data that adheres more closely to domain knowledge, enhancing its practical utility and fidelity.

Table 2: SA and MLE scores to evaluate authenticity. $\uparrow$ ($\downarrow$) indicates that the higher (lower) the score, the better the performance. The best results are highlighted in bold, and suboptimal ones are marked with an underline. The Average Gap measures the gap between the generated and real data.

| Data | Metrics | Real | Method | | | | | | | |
|---|---|---|---|---|---|---|---|---|---|---|
| | | | CTGAN | CTGAN+ | TVAE | P-TA | TabDDPM | TABSYN | TABDIFF | ECR-DM |
| Beijing | $\downarrow$RMSE | $0.421_{0.005}$ | $0.850_{0.054}$ | $0.960_{0.180}$ | $0.833_{0.058}$ | $1.324_{0.131}$ | $0.634_{1.331}$ | $0.613_{0.024}$ | $\underline{0.582_{0.012}}$ | $\mathbf{0.496_{0.006}}$ |
| | $\uparrow$SA | $100.00_{0.00}$ | $24.75_{18.48}$ | $87.84_{6.48}$ | $19.92_{3.38}$ | $39.02_{2.52}$ | $97.03_{26.15}$ | $97.58_{0.07}$ | $\underline{98.60_{0.30}}$ | $\mathbf{98.64_{0.07}}$ |
| News | $\downarrow$RMSE | $0.853_{0.005}$ | $0.860_{0.015}$ | $6.840_{0.030}$ | $0.993_{0.045}$ | $\underline{0.821_{0.011}}$ | $3.104_{0.000}$ | $0.846_{0.019}$ | $0.864_{0.021}$ | $\mathbf{0.814_{0.002}}$ |
| | $\uparrow$SA | $100.00_{0.00}$ | $0.00_{0.00}$ | $0.00_{0.00}$ | $0.00_{0.00}$ | $0.00_{0.00}$ | $0.00_{0.00}$ | $0.76_{0.04}$ | $\mathbf{1.42_{1.17}}$ | $\underline{1.20_{0.04}}$ |
| Shoppers | $\uparrow$AUC | $92.39_{0.23}$ | $87.95_{0.74}$ | $88.08_{1.41}$ | $88.48_{2.02}$ | $91.30_{0.54}$ | $81.35_{18.36}$ | $88.77_{0.76}$ | $\underline{92.27_{0.21}}$ | $\mathbf{92.92_{0.26}}$ |
| | $\uparrow$SA | $100.00_{0.00}$ | $5.28_{2.29}$ | $94.01_{3.93}$ | $10.80_{2.05}$ | $0.00_{0.00}$ | $0.00_{0.00}$ | $97.97_{0.10}$ | $\underline{99.19_{0.09}}$ | $\mathbf{99.97_{0.03}}$ |
| Adult | $\uparrow$AUC | $92.50_{0.25}$ | $89.15_{0.13}$ | $89.83_{0.07}$ | $88.66_{1.19}$ | $\mathbf{91.58_{0.15}}$ | $90.86_{0.38}$ | $89.65_{0.43}$ | $91.24_{0.08}$ | $\underline{91.32_{0.07}}$ |
| | $\uparrow$SA | $100.00_{0.00}$ | $0.00_{0.00}$ | $81.59_{0.91}$ | $0.00_{0.00}$ | $10.16_{17.59}$ | $\underline{92.66_{0.17}}$ | $61.95_{0.29}$ | $91.59_{0.29}$ | $\mathbf{93.25_{0.79}}$ |
| Default | $\uparrow$AUC | $76.48_{0.13}$ | $73.50_{0.74}$ | $68.67_{7.47}$ | $72.55_{1.36}$ | $74.29_{0.17}$ | $\underline{75.94_{0.18}}$ | $74.97_{0.78}$ | $74.56_{0.39}$ | $\mathbf{77.15_{0.46}}$ |
| | $\uparrow$SA | $100.00_{0.00}$ | $52.56_{2.22}$ | $98.78_{1.19}$ | $73.54_{4.13}$ | $12.22_{21.17}$ | $99.29_{0.29}$ | $99.94_{0.02}$ | $\underline{99.97_{0.01}}$ | $\mathbf{100.00_{0.00}}$ |
| Magic | $\uparrow$AUC | $94.65_{0.21}$ | $82.97_{0.61}$ | $87.31_{0.45}$ | $90.12_{0.47}$ | $88.51_{0.59}$ | $93.21_{0.55}$ | $85.44_{0.20}$ | $\underline{93.55_{0.43}}$ | $\mathbf{93.65_{0.35}}$ |
| | $\uparrow$SA | $100.00_{0.00}$ | $87.75_{0.99}$ | $97.39_{0.80}$ | $98.56_{0.22}$ | $44.47_{0.06}$ | $99.27_{0.11}$ | $93.02_{0.05}$ | $\underline{99.28_{0.06}}$ | $\mathbf{99.78_{0.05}}$ |
| Average Gap | $\downarrow$MLE | - | 21.27% | 142.61% | 22.11% | 37.09% | 55.08% | 10.59% | $\underline{7.46\%}$ | $\mathbf{2.37\%}$ |
| | $\downarrow$SA | - | 71.61% | 23.40% | 66.20% | 82.35% | 19.03% | 24.80% | $\underline{18.32\%}$ | $\mathbf{17.86\%}$ |

Table 3: Shape and Trend scores to evaluate data fidelity.

| Data | Metrics | Method | | | | | | | |
|---|---|---|---|---|---|---|---|---|---|
| | | CTGAN | CTGAN+ | TVAE | P-TA | TabDDPM | TABSYN | TABDIFF | ECR-DM |
| Beijing | $\uparrow$Shape | $80.59_{1.18}$ | $89.30_{6.15}$ | $79.32_{1.40}$ | $88.07_{0.27}$ | $64.01_{30.05}$ | $85.24_{0.06}$ | $\mathbf{98.35_{0.28}}$ | $\underline{97.95_{0.15}}$ |
| | $\uparrow$Trend | $80.38_{0.62}$ | $77.96_{5.60}$ | $81.67_{1.68}$ | $76.44_{2.89}$ | $58.56_{31.97}$ | $76.32_{0.03}$ | $96.86_{0.23}$ | $\mathbf{97.05_{0.03}}$ |
| News | $\uparrow$Shape | $83.86_{0.66}$ | $49.56_{1.74}$ | $83.07_{0.39}$ | $\mathbf{96.90_{0.03}}$ | $13.52_{6.21}$ | $96.45_{0.07}$ | $96.62_{0.29}$ | $\mathbf{96.90_{0.31}}$ |
| | $\uparrow$Trend | $94.82_{0.09}$ | $89.04_{0.43}$ | $93.41_{0.03}$ | $88.49_{0.10}$ | $28.45_{47.59}$ | $\underline{98.35_{0.03}}$ | $98.34_{0.32}$ | $\mathbf{98.60_{0.18}}$ |
| Shoppers | $\uparrow$Shape | $77.42_{1.78}$ | $70.00_{5.41}$ | $76.59_{1.23}$ | $83.35_{0.01}$ | $97.11_{0.64}$ | $94.90_{0.03}$ | $\mathbf{98.14_{0.21}}$ | $\underline{97.19_{0.33}}$ |
| | $\uparrow$Trend | $86.78_{0.21}$ | $70.52_{3.27}$ | $81.88_{1.44}$ | $52.75_{0.10}$ | $93.87_{1.67}$ | $92.40_{0.30}$ | $\mathbf{97.88_{0.15}}$ | $\underline{96.79_{0.35}}$ |
| Adult | $\uparrow$Shape | $83.78_{1.89}$ | $83.34_{5.94}$ | $85.10_{0.91}$ | $91.20_{0.18}$ | $98.94_{0.13}$ | $95.12_{0.02}$ | $98.84_{0.38}$ | $\mathbf{98.99_{0.10}}$ |
| | $\uparrow$Trend | $83.22_{2.54}$ | $82.11_{8.21}$ | $85.68_{1.41}$ | $76.81_{3.07}$ | $\mathbf{97.75_{0.52}}$ | $86.73_{0.65}$ | $97.50_{0.56}$ | $\underline{97.66_{0.46}}$ |
| Default | $\uparrow$Shape | $86.14_{0.39}$ | $85.56_{0.66}$ | $88.92_{0.33}$ | $93.36_{0.04}$ | $98.35_{0.10}$ | $96.57_{0.09}$ | $\underline{98.37_{0.37}}$ | $\mathbf{98.46_{0.18}}$ |
| | $\uparrow$Trend | $76.70_{0.83}$ | $82.42_{1.55}$ | $80.08_{2.75}$ | $27.92_{0.93}$ | $94.13_{0.26}$ | $87.56_{1.55}$ | $\underline{96.46_{1.26}}$ | $\mathbf{97.60_{0.26}}$ |
| Magic | $\uparrow$Shape | $89.70_{2.19}$ | $76.87_{0.83}$ | $91.43_{0.37}$ | $87.74_{0.26}$ | $\mathbf{98.77_{0.19}}$ | $92.31_{0.11}$ | $98.64_{0.20}$ | $\underline{98.70_{0.42}}$ |
| | $\uparrow$Trend | $89.38_{0.99}$ | $88.30_{0.39}$ | $93.87_{0.63}$ | $83.17_{0.03}$ | $\mathbf{99.17_{0.23}}$ | $91.10_{0.05}$ | $98.66_{0.22}$ | $\underline{98.86_{0.56}}$ |

### 5.2.2 FIDELITY EVALUATION

To evaluate ECR-DM in terms of replicating the distribution of real-world data while capturing inter-column dependencies (Q2), we conduct experiments in Table 3 and addition experiments in Appendix D Table [6-8] with the following two key observations.

**ECR-DM can faithfully replicate the real data distribution.** As shown in Table [3,6-8], ECR-DM outperforms the current best baseline method, TABDIFF, across all distribution evaluation metrics, demonstrating its superior ability to replicate the distribution of real data. The higher Shape score indicates that ECR-DM better captures intra-column distributions, while the improved $\alpha$-Precision and $\beta$-Recall show that ECR-DM preserves both fine-grained details and overall distribution consistency (Detailed in Appendix D.1). Additionally, the higher C2ST score suggests that ECR-DM more effectively preserves the real data pattern (Detailed in Appendix D.2).

**ECR-DM can maintain inter-column dependencies.** Table 3 shows that ECR-DM achieves optimal or near-optimal performance in metrics sensitive to column correlations, such as Trend and Shape. These results indicate that ECR-DM successfully captures the relationships between columns, maintaining realistic inter-column dependencies and generating synthetic data that aligns closely with real-world patterns.

### 5.3 COMPONENT ANALYSIS AND DATA VISUALIZATION

### 5.3.1 ABLATION STUDIES

To evaluate the contribution of each individual component in ECR-DM to the overall model performance (Q3), we conduct experiments in Table 4 with the following two key observations.

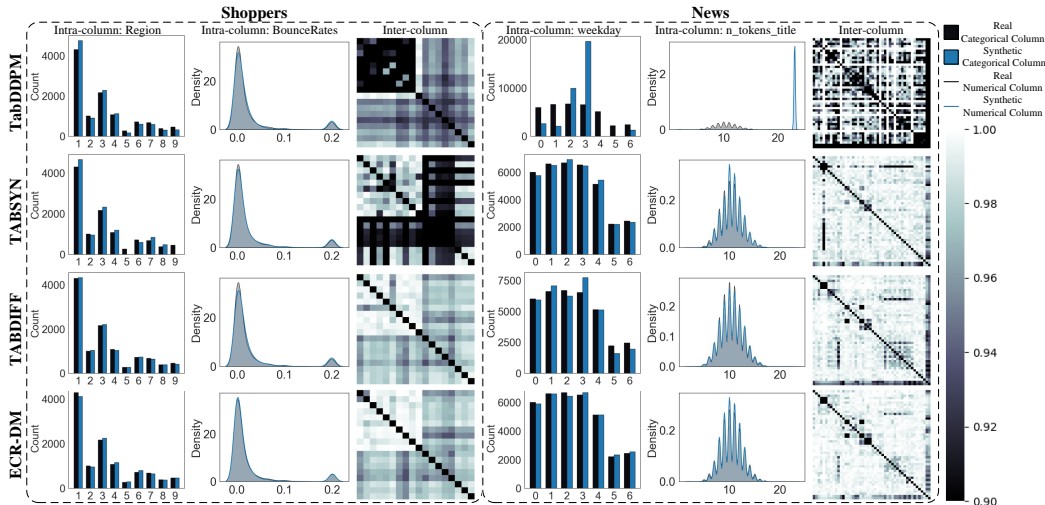

Figure 3: Visualize of both intra- and inter-column distributions.

**PLM encoder alone provides limited improvement.** Replacing the MLP encoder with a pretrained PLM encoder yields only marginal gains in both AUC and SA, as the PLM mainly projects inputs into a unified semantic space without capturing the full complexity of tabular data.

**Column-level diffusion and constraints are critical.** Adding column-level noise and constraint guidance significantly improves both AUC and SA. The combination of all modules, including PLM, column-level diffusion(CLD), constraint injection (CI), and column-name guidance (CLG), achieves the highest AUC and full SA, demonstrating that these components are essential for generating high-quality synthetic data that preserves inter-column dependencies and adheres to domain knowledge.

Table 4: Effect of component

| Encoder | CLD | CI | CLG | AUC | SA |
|---------|-----|----|-----|-----|-----|
| MLP | × | × | × | 75.11 | 99.94 |
| PLM | × | × | × | 75.23 | 99.95 |
| PLM | ✓ | × | × | 76.31 | 99.98 |
| PLM | × | ✓ | × | 76.80 | 99.97 |
| PLM | × | × | ✓ | 75.32 | 99.95 |
| PLM | × | ✓ | ✓ | 76.89 | 99.98 |
| PLM | ✓ | ✓ | ✓ | 77.15 | 100.00 |

### 5.3.2 SYNTHETIC DATA VISUALIZATION

To further analyze the effectiveness of ECR-DM, we visualize the generated synthetic data on the Shoppers and News datasets in Fig. 3 and Fig. [6-8], with the following two key observations:

**ECR-DM preserves real column distributions.** Visualizations of categorical columns (e.g., Region in Shoppers and Weekday in News) and numerical columns (e.g., BounceRates in Shoppers and n_tokens_title in News) show that ECR-DM closely matches the real data distributions. Compared with other diffusion-based methods such as TabDDPM, TABSYN, and TABDIFF, ECR-DM better preserves the distributions of minority classes (Detailed in Appendix E.1) and accurately captures peaks in numerical columns (Detailed in Appendix E.2). These results indicate that the Noise Perturbation Mechanism helps ECR-DM focus on fine-grained details, producing synthetic data that is more realistic and aligned with real-world patterns.

**ECR-DM captures inter-column dependencies.** Heatmaps of pairwise column correlations in the Shoppers and News datasets show that ECR-DM maintains stronger correlation structures compared with baseline diffusion methods, which exhibit more disrupted patterns (Detailed in Appendix E.3). This demonstrates that ECR-DM effectively models inter-column relationships, ensuring that synthetic samples reflect realistic dependencies across columns. Refer to Appendix E for more detailed distribution visualizations and Appendix D.4 for authenticity visualizations.

### 6 CONCLUSION

This paper introduces ECR-DM, a novel Explicit Column Relationship-Based Diffusion Model for synthetic tabular data generation that explicitly captures inter-column dependencies, learns the real data distribution, and preserves real-world tabular constraints. The key innovation of ECR-DM lies in integrating real-world tabular constraints into a carefully designed diffusion model, which explicitly guides the capture of inter-column dependencies. We validate ECR-DM through extensive experiments on six tabular data benchmarks, achieving superior performance across seven evaluation metrics, especially on downstream tasks.

ETHICS STATEMENT

This research is dedicated solely to scientific inquiry without involving human subjects, animals, or materials that may pose environmental concerns. As such, we do not anticipate any ethical risks or conflicts of interest. We are committed to upholding the highest standards of scientific integrity and ethics to ensure the accuracy and credibility of our findings.

REPRODUCIBILITY STATEMENT

To ensure the reproducibility of our work, we provide detailed descriptions of all components of our method, datasets, and experimental setups. The main paper includes the algorithmic description of our proposed ECR-DM model (Section 4) and an overview of the training and synthetic tabular data generation procedures (Appendix B). Additional implementation details and hyperparameter settings are provided in Section 5.1. Statistical summaries of the tabular datasets used in our experiments are described in Appendix C.1. All evaluation metrics is detailed in Section 5.1 and Appendix C.3. To further facilitate reproducibility, the source code is available through an anonymous downloadable repository: https://anonymous.4open.science/r/ECR-DM-0C72. Together, these materials provide the necessary information to reproduce the results reported in this paper.

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

## A    USE OF LLMs

LLMs were employed in this work to assist in writing and polishing the paper. All substantive research content, results, and analyses were independently conducted by the authors. The use of LLMs was limited to improving clarity, grammar, and presentation, and did not influence any scientific findings. Detailed experimental methods and results are fully described in the main text of the paper.

## B    ALGORITHMS

---

**Algorithm 1** Training of ECR-DM

---

1: **repeat**
2:   Sample $x \sim p(x)$
3:   **for** $i = 1, \ldots, n$ **do**
4:     $h_i = \text{PLM}(x_i, f_i)$ (Eq. 1)
5:     // NPM
6:     $h_i^t = h_i^0 + \sigma_i(t)\varepsilon, \varepsilon \sim \mathcal{N}(0, I)$ (Eq. 2)
7:     // CGR $s_\theta$
8:     $h_i^t \leftarrow \text{Time Embedding}(t)$
9:     $\tilde{h}_i^t = \text{Tabular Constrained Injection}(h_i^t, C)$ (Eq. 8)
10:     $\overline{h}_i^t = \text{Noise Column Annotation}(\tilde{h}_i^t, f_i)$ (Eq. 9)
11:   **end for**
12:   $\hat{h}_1^t, \ldots, \hat{h}_n^t = \text{Dependencies Capture}(\overline{h}_1^t, \ldots, \overline{h}_n^t)$ (Eq. 10)
13:   $\hat{\epsilon}_i = \text{MLP}(\hat{h}_i^t)$
14:   Calculate denoising loss: $\mathcal{L}_d = \sum_{i=1}^{n} \|\hat{\epsilon}_i - \epsilon_i\|_2^2, \epsilon_i = \sigma_i(t)\varepsilon$
15:   Update parameters $\theta$ using Adam optimizer
16: **until** converges

---

**Algorithm 2** Synthetic tabular data generation of ECR-DM

---

1: // Initialize sampling process for each column
2: **for** $i = 1, \ldots, n$ **do**
3:   Sample $h_i^T \sim \mathcal{N}(0, \sigma_i^2(T)I), t_{\max} = T$
4: **end for**
5: $h^T = [h_1^T, \ldots, h_n^T]$
6: // CGR
7: **for** $j = \max, \ldots, 1$ **do**
8:   **for** $h_i^{t_j}$ in $h^{t_j}$ **do**
9:     $\nabla_{h_i^{t_j}} \log p_i(h_i^{t_j}) \approx -s_\theta(h_i^{t_j}, t_j, C, f_i, h_1^{t_j}, \ldots, h_n^{t_j})/\sigma_i(t_j)$ (Eq. 7)
10:     Get $h_i^{t_{j-1}}$ via CGR in Eq. 5
11:   **end for**
12: **end for**
13: $h' = [h_1^{t_0}, \ldots, h_n^{t_0}]$
14: // Tabular Reconstruction
15: Put $h'$ as input of the Tabular Reconstruction in Eq. 13, then acquire Synthetic sample $x'$
16: $x' \in D'$ is the sampled synthetic tabular data

---

In this section, we present the training and data generation algorithms for ECR-DM, as detailed in Algorithm 1 and Algorithm 2, respectively.

The training of ECR-DM begins by sampling data $x$ from the true data distribution $p(x)$. For each column, the feature $x_i$ is encoded using a PLM. The NPM is then applied to each encoded column feature $h_i$, adding noise at time $t$ according to a noise column-specific schedule $\sigma_i(t)$. Then in CGR $s_\theta$ the resulting noisy features $h_i^t$ are further injected with real-world tabular constraints $C$ using the Tabular Constrained Injection, as shown in Eq. 8, and then annotated the real-world

tabular constrained noisy features $\tilde{h}_i^t$ using Noise Column Annotation, as shown in Eq. 9. Next, the inter-column dependencies among the column name annotated real-world tabular constrained noisy features $\overline{h}_i^t$ are captured through the Dependencies Capture module, as described in Eq. 10. The model finally predicts the noise $\hat{\epsilon}_i$ using a MLP. The denoising loss $\mathcal{L}_d$ is computed as the difference between the predicted and true noise, and the model parameters $\theta$ are updated using the Adam optimizer. This process continues until convergence.

For the synthetic tabular data generation, ECR-DM begins by sampling initial noise values for each column, denoted as $h_i^T$, from a normal distribution with column-specific variance. The algorithm then performs a CGR $s_\theta$, which is trained in Algorithm 1, to iteratively refine the noise. At each time step $t_j$, the reverse process incorporates real-world tabular constraints and domain knowledge. This process continues until the final values $h'$ are obtained. In the final step, the tabular reconstruction transforms $h'$ into a synthetic tabular data sample $x'$ that adheres to the real data distribution, inter-column dependencies, and practical constraints.

It is worth noting that the PLM and Tabular Reconstruction models in ECR-DM are pre-trained using the method from Zhang et al. (2024), with their parameters frozen during the training of ECR-DM.

## C  DETAILED EXPERIMENT SETUPS

### C.1  DATASETS

Table 5: Statistics of datasets. # Num/Cat stands for the number of numerical columns and the number of categorical columns, respectively. # Inter-/Intra-Column stands for the number of Inter-Column constraints and Intra-Column constraints, respectively.

| Dataset | # Rows | # Num/Cat | # Train/Validation/Test | # Inter-/Intra-Colunm | Task |
|---------|--------|-----------|-------------------------|-----------------------|------|
| Adult | 48842 | 6/9 | 28943/3618/16281 | 3/13 | Classification |
| Default | 30000 | 14/11 | 24000/3000/3000 | 3/24 | Classification |
| Shoppers | 12330 | 10/8 | 9864/1233/1233 | 5/17 | Classification |
| Magic | 19019 | 10/1 | 15215/1902/1902 | 5/10 | Classification |
| Beijing | 43824 | 7/5 | 35058/4383/4383 | 5/11 | Regression |
| News | 39644 | 46/2 | 31714/3965/3965 | 10/47 | Regression |

We use six tabular datasets from the UCI machine learning library[2], including Adult, Default, Shoppers, and Magic datasets for classification tasks, as well as Beijing and News datasets for regression tasks. The statistical information of the dataset is shown in Table 5. Real-world tabular constraints for each dataset are shown in Tables [9-14].

### C.2  BASELINES

In this section, we introduce the baseline methods used in this paper in three groups:

1) Traditional methods include GAN-based methods (e.g., CTGAN and CTGAN+) and VAE-based methods (e.g., TVAE):

- CTGAN and TVAE are two methods for synthetic tabular data generation proposed by Xu et al. (2019). While both methods share the same underlying framework, they are built on different generative models: CTGAN is based on GANs, whereas TVAE relies on VAEs. Both approaches incorporate two key components: (1) mode-specific normalization, designed to handle numerical columns with complex distributions; and (2) conditional generation of numerical columns based on categorical columns, aimed at addressing class imbalance issues.
- CTAB-GAN+ (Zhao et al., 2024) builds upon existing methods CTGAN by incorporating RDP-based privacy accounting, similar to DP-WGAN (Huang et al., 2022). Furthermore, by leveraging the Was+GP loss, CTAB-GAN+ effectively constrains the gradient norm,

---

[2]https://archive.ics.uci.edu/datasets

eliminating the need for weight clipping and resulting in more stable training for differentially private GANs.

2) LLM-based methods:

- P-TA (Yang et al., 2024) proposes the use of Proximal Policy Optimization (PPO (Schulman et al., 2017)) to apply GANs to guide LLMs to refine the probability distribution of tabular features and thereby generate more realistic data.

3) Diffusion model methods:

- TabDDPM (Kotelnikov et al., 2023) addresses the difficulty of handling categorical features in diffusion models by introducing additional categorical diffusion models specifically for categorical features. Despite its simplicity, our experiments have shown that TabDDPM achieves excellent performance.
- TABSYN (Zhang et al., 2024) synthesizes tabular data by leveraging a diffusion model within a VAE crafted latent space. Meanwhile, it adopts a simplified forward diffusion process, which adds Gaussian noises of linear standard deviation with respect to time, thus improving sampling speed.
- TABDIFF (Shi et al., 2025a) introduces a joint continuous-time diffusion process for both numerical and categorical data, enabling the modeling of multimodal distributions of tabular data within a single unified framework. To address the large differences in the distributions of individual features, TABDIFF employs a feature-based learnable diffusion process, which improves the model's ability to accurately capture the overall data distribution.

### C.3 METRICS

In this section, we provide a detailed introduction to all the evaluation metrics used in this paper.

#### C.3.1 SHAPE AND TREND

**Shape.** This metric measures the column-wise density estimation performance, which includes the Kolmogorov-Smirnov Test (KST) (Berger & Zhou, 2014) for numerical features and the Total Variation Distance (TVD) (Tao et al., 2024) for categorical data. Given two numerical distributions, $p_r(x)$ (representing real data) and $p_s(x)$ (representing synthetic data), KST quantifies the distance between these distributions by calculating the maximum discrepancy between their corresponding Cumulative Distribution Functions (CDFs):

$$\text{KST} = \sup_x |F_r(x) - F_s(x)|, \tag{14}$$

where $F_r(x)$ and $F_s(x)$ are the CDFs of $p_r(x)$ and $p_s(x)$, respectively:

$$F(x) = \int_{-\infty}^{x} p(x) \, dx. \tag{15}$$

For categorical data, the TVD is commonly used. TVD computes the frequency of each category and expresses it as a probability. The TVD score represents the average difference between the probabilities of each category:

$$\text{TVD} = \frac{1}{2} \sum_{\omega \in \Omega} |R(\omega) - S(\omega)|, \tag{16}$$

where $\omega$ represents all possible categories in a given column $\Omega$, and $R(\cdot)$ and $S(\cdot)$ denote the real and synthetic frequencies for these categories, respectively.

**Trend.** This metric measures the pair-wise column correlation estimation performance, which includes the Pearson Score for a pair of numerical columns and the Contingency Score for a pair of

categorical columns:

$$
\text{Pearson Score} = \frac{1}{2}\mathbb{E}_{x,y}\left|\rho^R(x,y) - \rho^S(x,y)\right|,
$$
$$
\text{Contingency Score} = \frac{1}{2}\sum_{\alpha \in A}\sum_{\beta \in B}\left|R_{\alpha,\beta} - S_{\alpha,\beta}\right|,
$$

(17)

where $\rho^R(x,y)$ and $\rho^S(x,y)$ denotes the Pearson correlation coefficient between column $x$ and column $y$ of the real data and synthetic data, respectively. $\alpha$ and $\beta$ describe all the possible categories in column A and column B, respectively. $R_{\alpha,\beta}$ and $S_{\alpha,\beta}$ are the joint frequency of $\alpha$ and $\beta$ in the real data and synthetic data, respectively.

### C.3.2 $\alpha$-PRECISION AND $\beta$-RECALL

Following Zhang et al. (2024), we adopt the $\alpha$-Precision and $\beta$-Recall metrics introduced in Alaa et al. (2022) to evaluate the quality of synthetic tabular data. These two metrics are designed to provide sample-level assessments of how well the synthetic data approximates the real data distribution, focusing on two critical aspects: fidelity and coverage.

$\alpha$-Precision quantifies the fidelity of the synthetic data. Specifically, it measures the proportion of synthetic samples that lie within the support of the real data distribution. In essence, it evaluates whether the synthetic data points are realistic and indistinguishable from genuine data samples. A high $\alpha$-Precision score indicates that the synthetic generator avoids producing out-of-distribution samples. On the other hand, $\beta$-Recall measures the coverage of the real data by the synthetic data. It reflects how well the synthetic samples represent the entire variability of the real data. In practice, $\beta$-Recall evaluates whether every real data point is "close enough" to at least one synthetic sample, thereby assessing how comprehensively the synthetic distribution captures the diversity present in the original dataset.

Together, $\alpha$-Precision and $\beta$-Recall provide a balanced view of synthetic data quality: the former ensures fidelity, while the latter ensures completeness. These metrics are particularly important for applications such as data augmentation, privacy-preserving data sharing, and model training, where both overfitting to narrow patterns and missing important data characteristics can significantly harm downstream performance.

### C.3.3 MACHINE LEARNING EFFICIENCY (MLE)

To measure the ability of synthetic tabular data to support downstream task learning, we evaluate their performance using Machine Learning Efficiency (MLE). Specifically, we adopt the Training on Synthetic and Testing on Real (TSTR) (Fekri et al., 2019) scheme. In this approach, the real dataset is first split into a training set and a testing set. Next, a generation model is used to synthesize synthetic data that matches the size of the training set. This synthetic data is then used to train an XGBoost classifier or XGBoost regressor. Finally, we evaluate these machine learning models based on the real test set, calculating the AUC scores for classification tasks and the RMSE for regression tasks, respectively.

### C.3.4 ABILITY TO ACCURATELY GENERATE SAMPLE

Sample Accuracy (SA) evaluates the ability of a generative model to correctly generate samples, that is, whether the generated samples can accurately follow real-world tabular constraints and domain knowledge, as shown in Tables [9-14]. Enhancing the model's ability to generate accurate and representative samples can significantly reduce the likelihood of producing biased or even toxic outputs. This improvement is particularly important for downstream tasks, where the quality and reliability of the synthetic data directly impact model performance, fairness, and safety.

### C.3.5 Detection

Detection evaluates how difficult it is to distinguish synthetic data from real data when the two are mixed. Specifically, we adopt the Classifier Two-Sample Test (C2ST) as implemented in the SDMetrics library[3], where a logistic regression model serves as the discriminator.

## D Addition Experimental Results

### D.1 Joint Distribution

Table 6: Comparison of $\alpha$-Precision scores. Bold Face represents the best score on each dataset and marks the suboptimal ones with an underline.

| Methods | Adult | Default | Shoppers | Magic | Beijing | News | Average | Ranking |
|---|---|---|---|---|---|---|---|---|
| CTGAN | $78.61_{1.85}$ | $68.48_{0.39}$ | $78.42_{3.32}$ | $83.25_{0.91}$ | $96.12_{1.48}$ | $97.00_{1.03}$ | 83.65 | 5 |
| CTGAN+ | $93.35_{2.68}$ | $87.57_{4.35}$ | $92.73_{2.87}$ | $41.74_{0.78}$ | $89.27_{11.23}$ | $0.00_{0.00}$ | 67.45 | 7 |
| TVAE | $96.71_{1.67}$ | $84.11_{1.76}$ | $62.29_{7.40}$ | $82.90_{0.92}$ | $90.58_{5.51}$ | $89.40_{8.92}$ | 84.33 | 4 |
| TabDDPM | $96.07_{0.34}$ | $97.73_{0.43}$ | $92.53_{1.74}$ | $98.20_{0.73}$ | $\mathbf{98.67}_{0.94}$ | $0.00_{0.00}$ | 69.66 | 6 |
| TABSYN | $94.34_{1.36}$ | $\mathbf{98.96}_{0.10}$ | $95.09_{0.40}$ | $90.93_{0.19}$ | $91.11_{0.41}$ | $\mathbf{99.30}_{0.05}$ | 94.95 | 3 |
| TABDIFF | $\underline{99.41}_{0.34}$ | $98.38_{0.47}$ | $\underline{99.28}_{0.49}$ | $\mathbf{99.32}_{0.34}$ | $98.59_{0.10}$ | $92.83_{2.25}$ | $\underline{97.97}$ | 2 |
| ECR-DM | $\mathbf{99.55}_{0.14}$ | $\underline{98.62}_{0.20}$ | $\mathbf{99.41}_{0.13}$ | $99.02_{0.18}$ | $96.40_{0.24}$ | $\underline{98.38}_{1.11}$ | $\mathbf{98.56}$ | 1 |

Table 7: Comparison of $\beta$-Recall scores. Bold Face represents the best score on each dataset and marks the suboptimal ones with an underline.

| Methods | Adult | Default | Shoppers | Magic | Beijing | News | Average | Ranking |
|---|---|---|---|---|---|---|---|---|
| CTGAN | $30.24_{1.19}$ | $19.77_{0.64}$ | $32.77_{0.84}$ | $10.67_{1.35}$ | $40.17_{1.42}$ | $26.18_{1.02}$ | 26.63 | 6 |
| CTGAN+ | $29.64_{7.37}$ | $21.59_{11.59}$ | $17.23_{3.02}$ | $0.17_{0.02}$ | $41.65_{1.99}$ | $0.00_{0.00}$ | 18.38 | 7 |
| TVAE | $36.67_{1.08}$ | $21.49_{0.98}$ | $23.20_{2.65}$ | $32.47_{0.44}$ | $26.26_{2.29}$ | $27.78_{1.17}$ | 27.98 | 5 |
| TabDDPM | $48.72_{0.75}$ | $\underline{47.16}_{0.82}$ | $\mathbf{54.13}_{0.74}$ | $\mathbf{47.57}_{0.93}$ | $19.13_{0.87}$ | $0.00_{0.00}$ | 36.12 | 3 |
| TABSYN | $33.24_{0.33}$ | $39.54_{0.26}$ | $38.66_{0.94}$ | $17.22_{0.08}$ | $20.34_{0.15}$ | $\mathbf{44.47}_{0.04}$ | 32.25 | 4 |
| TABDIFF | $\mathbf{49.37}_{1.21}$ | $\mathbf{50.53}_{0.82}$ | $49.51_{1.56}$ | $\underline{46.85}_{0.79}$ | $\underline{58.06}_{0.13}$ | $37.32_{45.34}$ | $\underline{48.67}$ | 2 |
| ECR-DM | $\underline{48.90}_{0.84}$ | $45.83_{0.22}$ | $\underline{53.09}_{0.50}$ | $46.53_{0.57}$ | $\mathbf{60.50}_{0.43}$ | $\underline{45.34}_{0.28}$ | $\mathbf{50.03}$ | 1 |

Experiments in Section 5.2.2 use column-related distribution, including the column-wise density estimation and pairwise column correlation estimation (Table 3), to evaluate the fidelity of synthetic data generated from different models. However, these results are insufficient to evaluate the synthetic data's overall density estimation performance. Therefore, in this section, we adopt $\alpha$-Precision and $\beta$-Recall to evaluate the joint distribution of the synthetic data.

As shown in Tables 6 and 7, we compared the $\alpha$-Precision and $\beta$-Recall scores of the ECR-DM and baseline, respectively. ECR-DM achieved improvements of 0.60% and 2.72% in these two metrics compared to the suboptimal method TABDIFF (Shi et al., 2025a), indicating that ECR-DM maintains a balance between extensive data coverage and preserving fine-grained details, thus faithfully capturing the breadth and depth of the true data distribution.

### D.2 Detection Score (C2ST)

ECR-DM achieves either the best or second-best performance in terms of the Detection Score measured by the C2ST. This metric reflects how distinguishable the synthetic data is from the real data. The strong performance of ECR-DM under this metric suggests that the synthetic samples generated are highly realistic and closely align with the underlying real data distribution. Compared to baseline models, ECR-DM produces synthetic data that is more difficult for a classifier (logistic regression) to distinguish from real data, demonstrating its effectiveness in preserving key statistical properties while minimizing artifacts that often arise in generative processes. This property is particularly important for downstream applications where distributional consistency between synthetic and real data is crucial.

---

[3]https://docs.sdv.dev/sdmetrics

Table 8: Detection score (C2ST) using logistic regression classifier. Bold Face represents the best score on each dataset and marks the suboptimal ones with an underline.

| Methods | Adult | Default | Shoppers | Magic | Beijing | News |
|---|---|---|---|---|---|---|
| CTGAN | $0.644_{0.054}$ | $0.639_{0.025}$ | $0.711_{0.014}$ | $0.681_{0.014}$ | $0.774_{0.078}$ | $0.695_{0.059}$ |
| CTGAN+ | $0.436_{0.284}$ | $0.660_{0.095}$ | $0.002_{0.001}$ | $0.020_{0.020}$ | $0.697_{0.220}$ | $0.000_{0.000}$ |
| TVAE | $0.703_{0.057}$ | $0.545_{0.034}$ | $0.297_{0.020}$ | $0.821_{0.031}$ | $0.694_{0.058}$ | $0.441_{0.023}$ |
| TabDDPM | $0.960_{0.009}$ | $\mathbf{0.981}_{0.011}$ | $0.861_{0.006}$ | $\mathbf{0.984}_{0.012}$ | $0.350_{0.534}$ | $0.162_{0.227}$ |
| TABSYN | $0.572_{0.019}$ | $0.715_{0.004}$ | $0.680_{0.007}$ | $0.720_{0.003}$ | $0.671_{0.001}$ | $0.839_{0.006}$ |
| TABDIFF | $\underline{0.967}_{0.012}$ | $\underline{0.949}_{0.008}$ | $\mathbf{0.959}_{0.029}$ | $0.970_{0.007}$ | $\mathbf{0.960}_{0.007}$ | $0.925_{0.029}$ |
| ECR-DM | $\mathbf{0.969}_{0.012}$ | $\underline{0.949}_{0.016}$ | $\underline{0.916}_{0.026}$ | $0.972_{0.024}$ | $0.906_{0.012}$ | $\mathbf{0.949}_{0.014}$ |

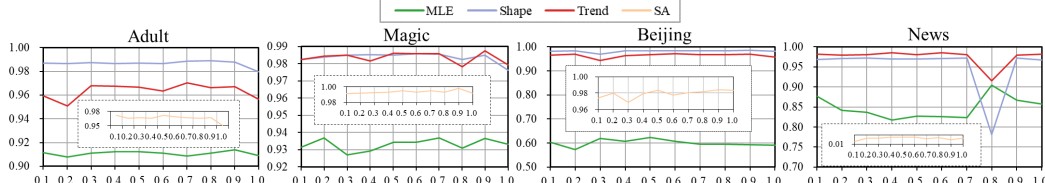

Figure 4: The impact of Condition Guidance Level Parameters $p_u$ on MLE, SA, Shape, and Trend across the Adult, Magic, Beijing, and News datasets.

### D.3 ANALYSIS OF CONDITION GUIDANCE LEVEL PARAMETERS

To verify the impact of different Condition Guidance Levels $p_u$ on the synthesized data (i.e., what extent condition information should be considered during the sampling process), we evaluate the variation trends of MLE, SA, Shape, and Trend with respect to the $p_u$ on four datasets: Adult, Magic, Beijing, and News. Adult and Magic are classification datasets, while Beijing and News are regression datasets.

As shown in Fig. 4, we observe that the MLE, SA, Shape, and Trend metrics exhibit similar trends in response to changes in the Condition Guidance Level Parameter $p_u$. The almost consistent pattern of changes in these metrics suggests that there may be a correlation between them, and that adjusting $p_u$ influences multiple aspects of the synthesized data in a related manner.

### D.4 VISUALIZATION OF AUTHENTICITY COMPARISON BETWEEN SYNTHETIC DATA AND REAL DATA

As shown in Fig. 5, we visualize the real Adult dataset alongside samples generated by TABSIF and ECR-DM. In Fig. 5, columns with the same color in each row indicate violations of real-world tabular constraints. The real dataset strictly adheres to these constraints; for example, if the "Relationship" column is "Husband," the "Gender" column must be "Male." While TABDIFF captures the overall data distribution, it ignores these constraints, resulting in unrealistic samples, such as assigning "Husband" to female entries. In contrast, ECR-DM integrates real-world constraints through the CGR process, improving its ability to generate data that respects constraints and substantially reducing the proportion of incorrect or misleading samples. This results in synthetic tabular data that is both realistic and practically useful.

## E DETAILED VISUALIZATION EXPERIMENT

This section supplements Section 5.3.2 by providing a more detailed description of the distribution visualization experiment, as outlined below:

**Real Data**

| age | workclass | fnlwgt | education | education.num | marital.status | occupation | relationship | race | sex | capital.gain | capital.loss | hours.per.week | native.country | income |
|---|---|---|---|---|---|---|---|---|---|---|---|---|---|---|
| 39 | State-gov | 77516 | Bachelors | 13 | Never-married | Adm-clerical | Not-in-family | White | Male | 2174 | 0 | 40 | United-States | <=50K |
| 38 | Private | 215646 | HS-grad | 9 | Divorced | Handlers-cleaners | Not-in-family | White | Male | 0 | 0 | 40 | United-States | <=50K |
| 50 | Self-emp-not-inc | 83311 | Bachelors | 13 | Married-civ-spouse | Exec-managerial | Husband | White | Male | 0 | 0 | 13 | United-States | <=50K |

**Synthetic Data of TABDIFF**

| age | workclass | fnlwgt | education | education.num | marital.status | occupation | relationship | race | sex | capital.gain | capital.loss | hours.per.week | native.country | income |
|---|---|---|---|---|---|---|---|---|---|---|---|---|---|---|
| 59 | Self-emp-inc | 71411.234 | HS-grad | 9 | Widowed | Sales | Not-in-family | White | Female | 8010.6533 | 0 | 40 | United-States | >50K |
| 51 | Private | 103360.13 | HS-grad | 8 | Widowed | Craft-repair | Husband | White | Female | 0 | 0 | 40 | United-States | >50K |
| 65 | Self-emp-inc | 167608.52 | 1st-4th | 1 | Never-married | Exec-managerial | Husband | White | Male | 7688 | 0 | 35 | United-States | >50K |

**Synthetic Data of ECR-DM**

| age | workclass | fnlwgt | education | education.num | marital.status | occupation | relationship | race | sex | capital.gain | capital.loss | hours.per.week | native.country | income |
|---|---|---|---|---|---|---|---|---|---|---|---|---|---|---|
| 50 | Self-emp-inc | 235320.89 | Some-college | 10 | Married-civ-spouse | Craft-repair | Husband | White | Male | 8231.633 | 0 | 60 | Germany | >50K |
| 53 | Local-gov | 189195.48 | Some-college | 10 | Married-civ-spouse | Exec-managerial | Wife | White | Female | 0 | 0 | 35 | United-States | >50K |
| 34 | Private | 332215.0 | 5th-6th | 2 | Married-civ-spouse | Adm-clerical | Wife | Asian-Pac-Islander | Female | 0 | 0 | 40 | Mexico | <=50K |

Figure 5: Visualize real and synthetic data of Adult dataset.

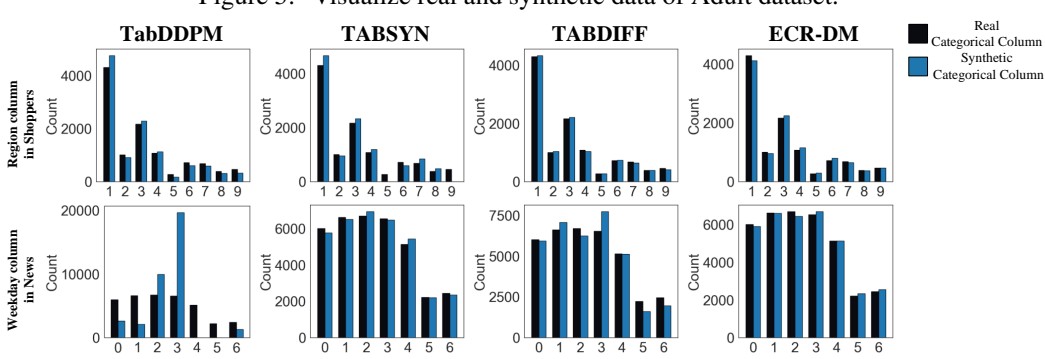

Figure 6: Visualization of Categorical Column Distribution.

## E.1 DETAILED DESCRIPTION OF CATEGORICAL COLUMN DISTRIBUTION VISUALIZATION

Fig. 6 visualizes the distribution of the categorical column Region in the Shoppers dataset and the categorical column Weekday in the News dataset. The horizontal axis represents the class of the categorical column, and the vertical axis represents the number of samples for each class in the dataset (i.e., real data and synthetic data generated by TabDDPM, TABSYN, TABDIFF, and ECR-DM, respectively).

As shown in Fig. 6, TabDDPM exhibits significant deviations from the real distribution for both the Region and Weekday columns. In contrast, ECR-DM better preserves the real distributions compared to TABSYN and TABDIFF. Notably, for the Region column in the Shoppers dataset, ECR-DM maintains the distributions of minority classes 5, 8, and 9 closely aligned with the real data, whereas TabDDPM and TABSYN show substantial deviations, and TABDIFF deviates significantly in class 9. These results indicate that ECR-DM effectively preserves the distributions of categorical columns and the NPM module enhances its ability to focus on fine-grained details, maintaining the distribution of minority class and producing more realistic synthetic data.

## E.2 DETAILED DESCRIPTION OF NUMERICAL COLUMN DISTRIBUTION VISUALIZATION

Fig. 7 visualizes the one-dimensional kernel density estimation (KDE) curves for the the numerical column BounceRates in the Shoppers dataset and the numerical column n_tokens_title in the News dataset, where the black line represents the real data and the blue line represents the synthetic data.

As shown in Fig. 7, TabDDPM, TABSYN, and TABDIFF poorly capture the peak values in the BounceRates column. In contrast, ECR-DM fits the BounceRates column more accurately. Moreover, ECR-DM also captures the distribution of more complex numerical columns, such as n_tokens_title, whereas TabDDPM exhibits a significant shift due to its limited modeling capac-

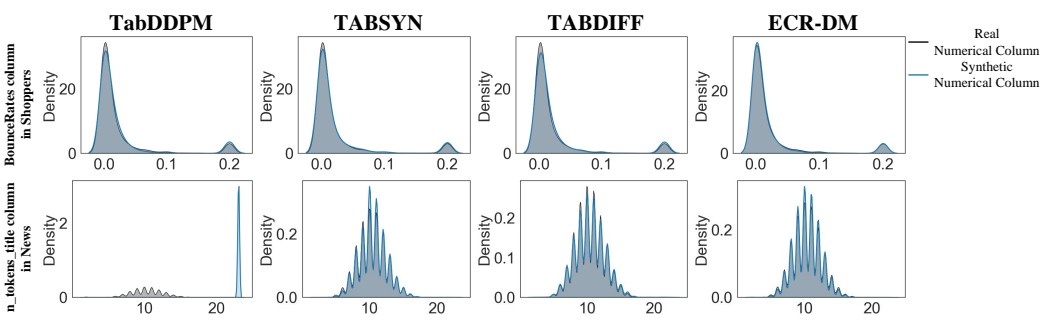

Figure 7: Visualization of inter-column distributions.

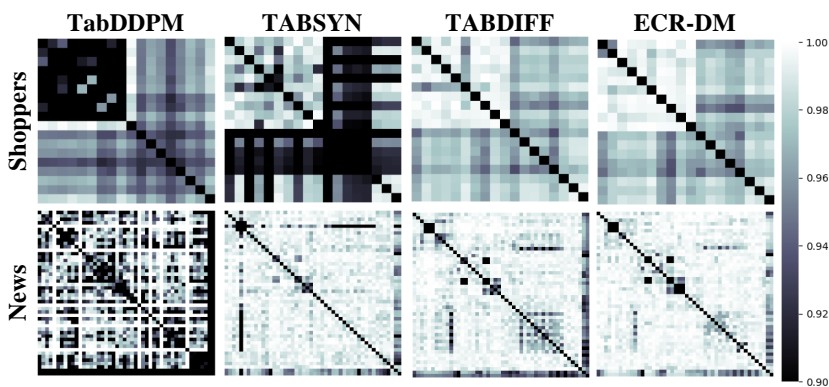

Figure 8: Visualization of inter-column distributions.

ity. These results demonstrate that ECR-DM effectively preserves the distributions of numerical columns.

### E.3 DETAILED DESCRIPTION OF NUMERICAL COLUMN DISTRIBUTION VISUALIZATION

Fig. 8 visualizes the ability of the synthetic Shoppers and News datasets to preserve inter-column correlations using a heatmap, where lighter colors indicate stronger correlation preservation. TabD-DPM and TABSYNshow an increase in dark blocks on both datasets, indicating a weaker ability to maintain inter-column correlations. In contrast, ECR-DM demonstrates strong correlation preservation on both datasets, further confirming its effectiveness in capturing inter-column dependencies.

Table 9: Real-world Tabular Constraints of Adult.

**Dataset Constraints**
Predict whether the annual income of an individual exceeds $50K/yr based on census data. Also known as the "Census Income" dataset.

**Intra-Column Constraints**
1.Age is between 17 and 90 (inclusive of 17, exclusive of 91).
2.Workclass must be one of a specific list (e.g., "Private", "Federal-gov", etc.).
3.Fnlwgt (final weight) must be a non-negative number.
4.Education must be from a predefined list (e.g., "HS-grad", "Masters", etc.).
5.Education number (numeric representation) must be between 1 and 16.
6.Marital status must match one of the listed valid options.
7.Occupation must match a set list of roles.
8.Relationship must be a valid type (e.g., "Husband", "Own-child").
9.Race, Sex, and Income must be one of the specified categories.
10.Capital gain must be between 0 and 99,999.
11.Capital loss must be non-negative.
12.Hours worked per week must be between 1 and 99.
13.Native country must be one of the recognized entries.

**Inter-Column Constraints**
1.If the relationship is "Husband" or "Wife", then marital status must imply the person is or was married. If someone is "Never-married", they shouldn't be listed as "Husband" or "Wife".
2.if someone has a higher education degree (Masters, Doctorate, Prof-school), they must be at least 22 years old.
3.Ensures that if the workclass indicates self-employment, the relationship should not be "Own-child".

Table 10: Real-world Tabular Constraints of Shoppers.

**Dataset Constraints**
Of the 12,330 sessions in the dataset, 84.5% (10,422) were negative class samples that did not end with shopping, and the rest (1908) were positive class samples ending with shopping.

**Intra-Column Constraints**
1.Administrative, Informational, ProductRelated: must be non-negative integers within known upper limits (e.g.,Administrative $\leq$ 27, Informational $\leq$ 24, ProductRelated $\leq$ 705).
2.Durations: (e.g., Administrative_Duration) must be non-negative.
3.BounceRates and ExitRates: must be between 0 and 0.2.
4.PageValues and SpecialDay: must be $\geq$ 0, with SpecialDay $\leq$ 1.
5.Month: must be a valid month string (e.g., "May", "Nov").

**Inter-Column Constraints**
1.If a visit count (e.g., Administrative) is zero, the corresponding duration must also be zero.
2.If duration is greater than 0, then the corresponding visit count must be $>$ 0.
3.If BounceRates $>$ 0.2, then ProductRelated should be $\leq$ 10.
4.If a user viewed fewer than one product-related page, then PageValues must be 0.
5.If PageValues $>$ 0, then ProductRelated_Duration $>$ 0 must be true.

Table 11: Real-world Tabular Constraints of Default.

Dataset Constraints
The dataset consists of feature vectors belonging to 12,330 sessions. The dataset was formed so that each session would belong to a different user in a 1-year period to avoid any tendency to a specific campaign, special day, user profile, or period.

Intra-Column Constraints
1.LIMIT_BAL: Credit limit is between 10,000 and 1,000,000.
2.SEX: Coded as 1 (male) or 2 (female).
3.EDUCATION: Must be one of 0–6 (0 = unknown, 1 = graduate school, etc.).
4.MARRIAGE: Must be one of 0–3 (0 = unknown, 1 = married, etc.).
5.AGE: Between 18 and 100 years.
6.PAY_0 to PAY_6: Monthly payment statuses must be in the range [-2, 8]. These indicate repayment behavior (e.g., -1 = paid in full, 1 = one month delay, etc.).
7.BILL_AMT1 to BILL_AMT6: Monthly bill amounts fall within observed realistic ranges (can be negative, possibly indicating credit).
8.PAY_AMT1 to PAY_AMT6: Payments are between 0 and an upper bound (no negative payments).
9.default_payment_next_month: Must be 0 or 1 (0 = no default, 1 = default).

Inter-Column Constraints
1.For any month where the payment status (PAY_*) indicates a delay greater than 1 month, the corresponding bill amount (BILL_AMT*) must be non-negative (i.e., the person owed money)
2.If there is any positive payment in a month, the total of all bill amounts should be greater than or equal to the total payments.
3.If a person is 20 years or younger, their credit limit (LIMIT_BAL) must be zero or less (suggesting they should not have a credit limit).

Table 12: Real-world Tabular Constraints of Magic.

Dataset Constraints
Data are MC generated to simulate registration of high-energy gamma particles in an atmospheric Cherenkov telescope.

Intra-Column Constraints
1.All physical quantities (like Length, Width, Size, Dist) must be non-negative.
2.Concentration ratios (Conc, Conc1) must be between 0 and 1.
3.Asymmetry and moment features must fall within realistic range limits (e.g., -500 to 580).
4.Alpha (angle in degrees) must be between 0 and 90.
5.class must be either 'g' (gamma-ray signal) or 'h' (hadron background).

Inter-Column Constraints
1.The major axis (Length) must be $\geq$ minor axis (Width), as expected for an ellipse.
2.The intensity from the highest pixel (Conc1) must be $\leq$ the sum of the top 2 pixels (Conc), and both must be $\leq 1$.
3.If the image is highly concentrated (Conc $> 0.5$), the total intensity (Size) should be sufficiently large ($>1.5$). This helps avoid extreme concentration in very faint showers, which is likely noise.
4.If the ellipse is centered (Dist $< 50$) and symmetric (Asym $\approx 0$), then the orientation angle (Alpha) should be small ($< 30°$). This reflects physically aligned gamma-ray events.
5.If the asymmetry is large (absolute value $> 100$ mm), then the third moment along the major axis (M3Long) should also be large ($>|8|$ mm), reflecting skewed energy distributions in real air showers.

Table 13: Real-world Tabular Constraints of Beijing.

Dataset Constraints
This data set contains the PM2.5 data of the US Embassy in Beijing. Meanwhile, meteorological data from Beijing Capital International Airport are also included.

Intra-Column Constraints
1.Year should be between 2010 and 2014.
2.Month: between 1 and 12, Day: between 1 and 31, Hour: between 0 and 23.
3.PM2.5 concentration (pm2_5) must be non-negative.
4.Dew point (DEWP) must be $\geq$ -100.
5.Temperature must be between -50 and 60 °C.
6.Pressure (PRES) must be between 800 and 1100 hPa.
7.Wind direction (cbwd) must be one of ('SE', 'NW', 'cv', 'NE').
8.Wind speed (Iws) and other count columns (Is, Ir) must be non-negative.

Inter-Column Constraints
1.Ensures that a month like 2 4 6 9 11 doesn't have 31 days.
2.For February, it checks if the year is a leap year, and sets the maximum allowed day to 29 (leap) or 28 (non-leap).
3.The dew point (DEWP) should never exceed the actual temperature (TEMP).
4.If wind speed (Iws) is high ($>$10), the PM2.5 (pm2_5) level is expected to be below 800 (because wind typically disperses pollutants).
5.If there is no wind (wind speed (Iws) = 0), then the wind direction (cbwd) should be 'cv' (likely short for "calm/variable").

Table 14: Real-world Tabular Constraints of News.

Dataset Constraints
This dataset summarizes a heterogeneous set of features about articles published by Mashable in a period of two years. The goal is to predict the number of shares in social networks (popularity).

Intra-Column Constraints
1.Numeric Features (e.g., n_tokens_title, average_token_length) must be non-negative.
2.num_keywords: must be between 1 and 10.
3.LDA_00 to LDA_04: must be between 0 and 1.
4.title_sentiment_polarity, avg_negative_polarity, etc., must lie in valid sentiment ranges (e.g., [-1, 1]).
5.Sharerelated features (e.g., shares, self_reference_min_shares): must lie between 0 and 843300 (the observed max).

Inter-Column Constraints
1.Ensures that the sum of positive and negative word rates is approximately 1 (±0.01 tolerance).
2.when there is at least one keyword, the three keyword minimum/average features are each $\geq$ -1.
3.Enforces that the average of keyword metrics follows a logical order: kw_min_avg $\leq$ kw_avg_avg $\leq$ kw_max_avg.
4.Self-reference shares follow: min $\leq$ avg $\leq$ max.
5.Verifies that the sum of all LDA topic probabilities is approximately 1.
6.Ensures: min_positive $\leq$ avg_positive $\leq$ max_positive
7.Ensures: min_negative $\leq$ avg_negative $\leq$ max_negative
8.Ensures: abs_title_sentiment_polarity == abs(title_sentiment_polarity)
9.If global_rate_positive_words $>$ 0, then avg_positive_polarity must be $>$ 0.
10.If global_rate_negative_words $>$ 0, then avg_negative_polarity must be $<$ 0.

