# OpenReview forum: "Explicit Column Relationship-Based Diffusion Model for High-Quality Synthetic Tabular Data Generation"
_ICLR.cc/2026/Conference — Submitted to ICLR 2026_

### Official Review · Reviewer_GK8V · 2025-10-28

**Soundness:** 2
**Presentation:** 2
**Contribution:** 2
**Rating:** 2
**Confidence:** 3

**Summary:**

This paper proposes a diffusion-based synthetic tabular data generative model. The model has a few components added to improve the diffusion model, such as adding column-specific noise and providing data constraints.

**Strengths:**

- The motivation of the work is clear - how to make synthetic data follow real-world constraints.
- Projecting tabular data into a semantic space and aligning it with natural language representation of real-world data constraints is a novel idea

**Weaknesses:**

- This paper has several add-ons to the diffusion model. Not all components actually contributed to improving the performance.
- Transforming categorical columns or numerical columns into a unified semantic space is proposed in several previous works (e.g., https://arxiv.org/abs/2205.09328). This work fails to correctly cite.
- It is unclear how the columns’ interdependency can be well reflected by adding individual noise to each column. How does the learning distribution of each column effectively (with individual noise) lead to preserving the inter-column dependency? There is a logical gap. This is not clarified in the manuscript. Adding separate noises seems to be very empirical, rather than based on rationales.
- How to set the right noise for each column? Is it dependent on the column’s statistics?
- It is unclear how the real-world constraint C in natural language’s representation space is well aligned with the semantic representation of tabular data.
- It is unclear how to generate high-quality real-world constraints. What if such constraints or rules are not obviously visible and the rules are latent?

**Questions:**

see weakness

---

### Official Review · Reviewer_ypzg · 2025-10-30

**Soundness:** 2
**Presentation:** 2
**Contribution:** 2
**Rating:** 0
**Confidence:** 4

**Summary:**

The paper proposes an approach to synthetic tabular data generation. Explicit Column Relationship-Based Diffusion Model (ECR-DM) introduces in the forward direction  a noise perturbation mechanism to learn column distributions and in the reverse direction constraint-guided discovery to adhere to inter column dependencies. The approach is evaluated on four classification and two regression datasets and against a variety of baselines.

**Strengths:**

The strengths are:
- The paper tackles a practically important problem of modeling tabular data.
- There proposed model is evaluated on a number of datasets and against a number of baselines.

**Weaknesses:**

The main weakness are:
- I had difficulty understanding the model. The problem set up did not include any optimization problem. Diffusion models themselves were not introduced. The technical exposition was heuristic and difficult to follow.
- Critiques of alternative models lacked precision, rendering the contribution unclear.
- The experiments seem promising but the lack of technical clarity made it hard to come to solid conclusions about the nature of performance.

Minor:
- The acronyms NPM and CGR are not introduced before they are used.
- Figure 1 is not helpful. It is too small, and the dependences do not require a picture. They would be more concisely described in words. (This is essentially what the figure does.)
- "They focus primarily on distributions and overlook the real, intricate relationships between columns" What does this claim mean? What is "overlook"ing in this context? Is there a quantitative claim that could more effectively make the point?
- "How to construct..." This sentence is awkward. It appears to be missing a verb. How should we...How ought we...How can we...Also the sentence should probably end with a question mark.
- "they still focus primarily on data distribution and fail to capture the complex inter-column dependencies" Again, it feels like there is a more precise way to word this claim. I don't really understand what is mean by it in the current form.
- "Furthermore, they often overlook real-world constraints (e.g., Real-world Tabular Constraints of Fig. 1)" Are these different than the inter-column dependencies? How? Are real-world constraints different from Real-world Tabular Constraints?).
- "Due to the unique reconstruction
process of diffusion models, they can incorporate realistic constraints, giving them a distinct advantage in this field. " <--- The beginning of this paragraph appears to be about how diffusion models don't capture these constraints?
- The introduction is largely redundant with the abstract. There are more words, but they don't really ad much.
- "We propose the Explicit Column Relationship-Based Diffusion Model (ECR-DM), which explicitly captures ..." double explicit
- The contributions are also repetitive.
- I would consider putting the related work near the end rather than at the beginning. It is hard to understand what might or might not be related in an interesting way without knowing more about what are the technical innovations of the current work.
- In the problem definition, I would have expected to see some kind of loss to minimized? I don't understand the problem based on the statement.

**Questions:**

Please see weaknesses.

---

### Official Review · Reviewer_oCA6 · 2025-11-01

**Soundness:** 3
**Presentation:** 2
**Contribution:** 3
**Rating:** 6
**Confidence:** 3

**Summary:**

This paper proposes ECR-DM, a diffusion-based framework designed to generate realistic synthetic tabular data while preserving real-world constraints and inter-column dependencies. Unlike prior GAN or diffusion methods that mainly focus on statistical similarity, ECR-DM explicitly models column-wise relationships and incorporates domain-specific logical constraints through a constraint-guided recovery mechanism in the reverse diffusion process. Experiments on six real-world datasets show substantial improvements in constraint satisfaction (SA up to 98–99%) and downstream task performance compared to baselines such as CTGAN, TabDDPM, and TABDIFF.

**Strengths:**

The proposed approach is novel and well-motivated. It clearly improves both the logical consistency and fidelity of synthetic tabular data compared to existing baselines. The idea of integrating explicit constraints into the reverse diffusion process is elegant and practical, and the experimental results are convincing.

**Weaknesses:**

The model relies on predefined constraints, which limits general applicability. The computational cost and interpretability of the constraint-guided recovery step are not fully discussed.

**Questions:**

1. Could the constraint embeddings be learned jointly rather than predefined?
2. How does ECR-DM behave when constraints are incomplete or partially inconsistent?

**Details Of Ethics Concerns:**

no ethics concerns

---

### Official Review · Reviewer_Cfui · 2025-11-22

**Soundness:** 1
**Presentation:** 1
**Contribution:** 1
**Rating:** 0
**Confidence:** 4

**Summary:**

Please read weaknesses

**Strengths:**

Please read weaknesses

**Weaknesses:**

In the first page of the draft (lines 033 through 041) there is a table that is located one line higher than the beginning of the section without keeping the corresponding section space from main text/figures/tables hence violating ICLR 2026 template
I recommend desk rejection

**Questions:**

Please read weaknesses

---

### Meta-Review · Area_Chair_HG5R · 2025-12-18

**Summary:**

This paper proposes two main contributions to the field of tabular data diffusion models. The first idea is a column-specific forward noising process to recognise the different semantics of each columns. The second idea is to introduce conditional generation via guidance, where conditioning is done by specifying constraints for column interdependencies and embedding the natural language descriptions of these constraints in the denoiser network. Both elements require a pre-trained language model to build embeddings. Experiments presented benchmark tabular data generation tasks widely used in the literature, where the proposed model claims state-of-the-art.

I briefly read the paper, and I think that while the key idea is novel to the best of my knowledge (I've been working on tabular diffusion models), the paper presentation is unclear especially in terms of the design choices for the embedding methods (also there's little ablation study to support the design choice). For example:

- Eq (1): Average over what? Also what does $Average(PLM([f_i])) \times x_i$ mean (where the average is taken outside of the $\times x_i$ operation)?
- Eqs (8) - (10): Not clear to me about the motivation of the specific design choices. For example, why applying the $C$ matrix to construct $K, V$ only but not $Q$?

Due to the lack of explanations and ablations, it is unclear to me (and to some reviewers) about the key source of the improvement gains.

**Reviewer Concerns:**

The major concerns from the reviewers are:

- Unclear details about the conditional guidance approach, e.g., computational speed, during training how to generate constraints that aligns well with the column dependencies in training data, etc.
- Insufficient amount of ablation studies.
- Violation of the template -- too many \vspace being used and quite some paragraphs and figures are cramped together.

No rebuttal is provided for this submission.

**Reviewer Scores:**

Reviewers all recommended rejection.

No author rebuttal is provided.

---

### Decision · Program_Chairs · 2026-01-26

Reject